# Optimizing the Conditions of Metal Solidification with Vibration

**Olga Kudryashova** [1,*] , **Marina Khmeleva** [1] , **Pavel Danilov** [1], **Vladislav Dammer** [1], **Alexander Vorozhtsov** [1] **and Dmitry Eskin** [1,2]

[1] Faculty of Physics and Engineering, Tomsk State University, Tomsk 634050, Russia; khmelmg@gmail.com (M.K.); padanilov@gmail.com (P.D.); dammer.tomsk@yandex.ru (V.D.); abv1953@mail.ru (A.V.); dmitry.eskin@brunel.ac.uk (D.E.)

[2] Brunel Centre for Advanced Solidification Technology, Brunel University London, Uxbridge UB8 3PH, UK

\* Correspondence: olgakudr@inbox.ru; Tel.: +790-5924-5444

**Abstract:** Vibration treatment of solidifying metals results in improvement in the ingot structure. There is a need to study this process not only because of the practical potential of vibration treatment but also due to the lack of understanding the process. An important practical challenge is to find optimal conditions for liquid metal processing. In this paper, the authors consider a solidification process in the particular case of a cylindrical chill mold with vibration as a solution of the Stefan problem. An integral value of mechanical stresses in the melt during solidification is considered as an efficiency criterion of vibration treatment. A dependence of this value on the vibration frequency and amplitude is obtained through solving the Stefan problem numerically. The solution allows one to find the optimal vibration frequency and amplitude. We verified the numerical solution with experimental data obtained upon vibration treatment of aluminum melt under different conditions. The experimentally found optimal conditions for metal processing were similar to those proposed in theory, i.e., a vibration frequency of about 60 Hz and an amplitude of about 0.5 mm.

**Keywords:** vibration treatment; optimal conditions; solidifying melt; Stefan problem

## 1. Introduction

Vibration treatment of a solidifying melt is one of the known methods of influencing the casting quality of aluminum alloys through reducing the grain size and obtaining a more uniform microstructure. An amplitude up to 5 cm and a frequency up to 10 kHz can be achieved with mechanical vibrations. At a higher frequency, e.g., above 16 kHz, ultrasonic treatment of melts can be performed [1]. Ultrasonic melt treatment of an A390 hypereutectic Al–Si alloy is shown to enhance the homogeneity of the microstructure and to increase the yield strength of the as-cast alloy by12% [2].

There are numerous theoretical and experimental works devoted to the influence of acoustic and mechanical vibrations on liquid metals during solidification (see for example [1] and [3–9]). It has been shown that vibration can modify the microstructure by multiplication of solidification substrates [5–7]. Study of the mechanical properties of A356 alloy after vibration treatment with frequencies of 100 Hz and 150 Hz showed the tensile and yield strengths improved by 20% and 10%, respectively, but the use of vibration frequencies of 200 Hz caused the formation of a high porosity microstructure and caused major defects [8]. The multiplication of grains is achieved through mechanical and thermal fragmentation of dendritic crystals by elastic stresses, microflows (thermal and solutal convection in a liquid) and cavitation [9–14]. Generally, the refinement of crystalline grains and the increasing soundness of a casting lead to improved mechanical properties and quality of cast products.

A mathematical model describing the vibration effect on liquid metals as a function of cavitation and turbulent flows in the bulk melt and in the solid-liquid zone has been proposed in [15]. Numerous prior studies showed the positive effect of both phenomena (cavitation and turbulence in the melt bulk) on the quality of casting microstructures [1,3,5]. These conditions can be implemented under high-intensity processes with relatively large amplitudes (more than 1 cm) and frequencies (more than 60 Hz but less than the ultrasonic range). In this case, cavitation in the liquid metal may occur in the melt saturated with gases (e.g., hydrogen in aluminum) during the solidification ("pseudocavitation") or under vibration treatment.

It was experimentally found [16] that the effect of grain refinement increases up to a certain frequency depending on the liquid alloy properties with an increase in the pre-ultrasonic vibration frequency. A further increase in the frequency reduces the effect as compared to an optimal vibration frequency (about 50 Hz with an amplitude of 0.49 mm in [16]). In our earlier work [15], the optimal frequency for high-intensity vibration treatment of melts was explained by the simultaneous occurrence of pseudocavitation and turbulent flows. The work [17], investigated the crystal behaviors under vibration using a transparent NH4Cl-70%H2O alloy (frequency from 20 to 1 000 Hz and acceleration from 10 to 100 m/s$^2$). The optimal frequency of 50 Hz and acceleration of 100 m/s$^2$ were found when the grain refinement effect was strengthened.

However, even in a more general case with low-intensity vibration, a small amount of gases in a melt, and a lack of cavitation and turbulence, vibration treatment of a solidifying alloy can also lead to a positive result [2,5]. In this case, there should also be an optimal frequency and amplitude under which the improvement of casting microstructure is most pronounced.

The aim of this study is to examine the mechanisms for improving the as-cast microstructures by vibration during alloy solidification; and based on a numerical solution of the Stefan problem to determine the optimal conditions (frequency and amplitude) for the vibration treatment without cavitation and turbulence.

## 2. Mathematical Model of Metal Solidification in a Cylindrical Volume with Vibration as a Stefan Problem

In considering solidification processes, an important issue is the local distribution of temperature field characteristics for a crystallizing ingot, such as temperature gradients and solidification-front velocity until the end of solidification.

The oscillating melt affects the stress distributions in the mushy (semi-solid) region of an ingot and influences the conditions of crystal growth. Under vibration, the moving melt rinses off saturated solute layers around growing crystals, increasing heat transfer and contributing to the dendrite growth. At the same time, the transfer of solute elements by a melt and their accumulation in interdendritic spaces can lead to local re-melting of the solid phase and the separation of dendritic branches, i.e., dendrite fragmentation [18]. The rate of heat flow with vibration treatment increases due to the convective heat transfer.

### 2.1. Model Setup

Let us consider the cooling and solidification of liquid metal in a thick-walled cylindrical chill mold subjected to low-frequency mechanical vibration (Figure 1). Mold walls are a source of horizontal vibration. Initially, the mold is uniformly heated to a temperature $T_s$, and a molten metal in the mold has a temperature $T_0$ above the liquidus temperature $T_{m1}$. Then the "liquid metal–hill mold" system cools down. The solidification starts at mold walls where the melt temperature at some point in time $t_{col}$ becomes equal to $T_{m1}$. From this moment, the metal solidification starts and the solidification front moves from the mold walls to the cylinder axis. Solidification ends at a time $t_{froz}$. Then, the "solid metal–chill mold" system continues to cool down.

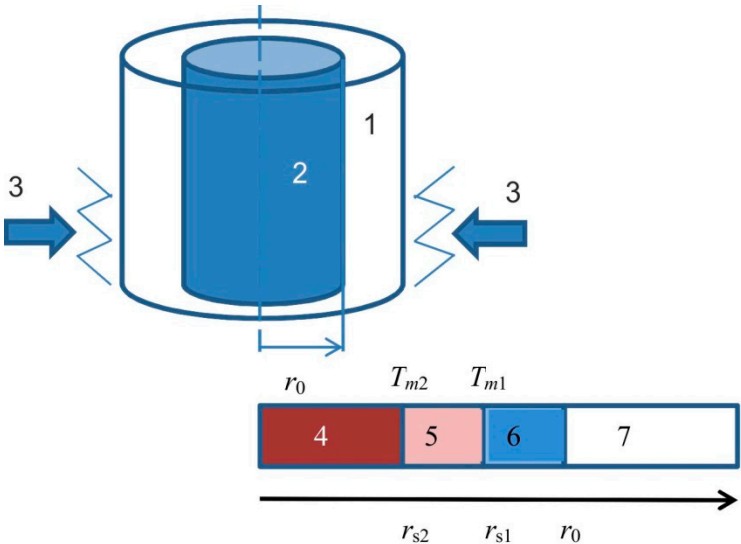

**Figure 1.** Scheme for the numerical setup of metal solidification with vibration: 1—thick-walled cylindrical chill mold, 2—liquid metal, 3—vibration generator, 4—molten zone, 5—transition zone, 6—zone of solid metal, 7—mold wall.

We made the following assumptions:

- The height of the chill mold is much larger than its inner diameter $2r_0$
- The heat transfer from the melt occurs only through the mold walls; the heat transfer at the top and bottom sides of the chill mold is ignored (an area of mold walls is much larger than the area of the mold bottom and open surface)
- The phase transition occurs within a temperature range from $T_{m1}$ (liquidus temperature) to $T_{m2}$ (solidus temperature)
- Microscopic volumes of the metal undergo longitudinal harmonic vibrations in an elastic wave with velocity $v$ depending on the vibration frequency and amplitude.

The mathematical model of the phase transition under these assumptions is limited to solving a one-dimensional Stefan problem in cylindrical coordinates [19,20].

$$
\begin{cases}
\dfrac{\partial T_1}{\partial t} + v\dfrac{\partial T_1}{\partial r} = \dfrac{a_1}{r}\dfrac{\partial}{\partial r}\left(r\dfrac{\partial T_1}{\partial r}\right), & r \le r_{s2} \\[2mm]
\dfrac{\partial T_{12}}{\partial t} = \dfrac{a_{12}(T)}{r}\dfrac{\partial}{\partial r}\left(r\dfrac{\partial T_{12}}{\partial r}\right), & r_{s1} > r > r_{s2} \\[2mm]
\dfrac{\partial T_2}{\partial t} = \dfrac{a_2}{r}\dfrac{\partial}{\partial r}\left(r\dfrac{\partial T_2}{\partial r}\right), & r_0 > r \ge r_{s1} \\[2mm]
\dfrac{\partial T_3}{\partial t} = \dfrac{a_3}{r}\dfrac{\partial}{\partial r}\left(r\dfrac{\partial T_3}{\partial r}\right), & r > r_0
\end{cases}
\tag{1}
$$

with initial and boundary conditions:

$$
\begin{aligned}
t = 0 &: r_{s1} = r_0, T = T_0 \text{ under } r < r_0; \ T = T_s \text{ under } r = r_0 \\
r = r_{s2} &: \frac{\partial T_1}{\partial r} = \frac{\partial T_{12}}{\partial r}, T_1 = T_{12} = T_{m2} \\
r = 0 &: \frac{\partial T}{\partial r} = 0; r = r_0 : \frac{\partial T_2}{\partial r} = \frac{\partial T_3}{\partial r}, T_2 = T_3 \\
r &\to \infty : T_3 = T_s
\end{aligned}
\tag{2}
$$

where $r$ is the cylindrical coordinate, $t$ is the time, $a = \frac{c\rho}{\lambda}$ is the thermal diffusivity, index 1 refers to the melt, index 2 to the solid phase, index 12 to the transition zone, and index 3 refers to the chill mold.

The effective coefficient of thermal diffusivity in the transition zone is defined as $a_{12}(T) = a_1 + f_s a_2$, where $f_s$ corresponds to the volume fraction of the solid phase in the transition (mushy) zone: $f_s = 1 - \frac{T-T_{m1}}{T_{m2}-T_{m1}}$.

The velocity of solidification front $\frac{\partial r_{s1}}{\partial t}$ is determined from the conditions at the border of solidification $r_{s1}$ (that is, corresponding to the liquidus temperature $T_{m1}$) [19–21]:

$$r = r_{s1} : \lambda_1 \frac{\partial T_1}{\partial r} - \lambda_2 \frac{\partial T_2}{\partial r} = \rho_1 L \frac{dr_{s1}}{dt}, \ T_1 = T_{12} = T_{m1}, \tag{3}$$

where $L$ is the latent heat of solidification, $\lambda_1$ and $\varrho_1$ are the thermal conductivity and liquid phase density, respectively, and $\lambda_2$ is the thermal conductivity of the solid phase.

Define an offset of fluid microscopic volumes in the environment by solving the set of wave equations:

$$\begin{cases} \frac{\partial^2 S_1}{\partial t^2} = \frac{c_1}{r} \frac{\partial}{\partial r} \left( r \frac{\partial S_1}{\partial r} \right), & r \leq r_{s2} \\ \frac{\partial^2 S_{12}}{\partial t^2} = \frac{c_{12}(T)}{r} \frac{\partial}{\partial r} \left( r \frac{\partial S_{12}}{\partial r} \right), & r_{s1} > r > r_{s2} \\ \frac{\partial^2 S_2}{\partial t^2} = \frac{c_2}{r} \frac{\partial}{\partial r} \left( r \frac{\partial S_2}{\partial r} \right), & r \geq r_{s1} \end{cases} \tag{4}$$

with initial and boundary conditions:

$$\begin{array}{c} t = 0 : r_s = r_0, S = 0; \\ r = r_{s2} : \frac{\partial S_1}{\partial r} = \frac{\partial S_{12}}{\partial r}, S_1 = S_{12}; \\ r = r_{s1} : \frac{\partial S_{12}}{\partial r} = \frac{\partial S_2}{\partial r}, S_{12} = S_2; \\ r = 0 : \frac{\partial S_1}{\partial r} = 0; r = r_0 : S = A \sin \omega t, \end{array} \tag{5}$$

where $S$ is the particle displacement relative to an equilibrium position, $c$ is the sound speed, $A$ and $\omega$ are the vibration amplitude and frequency, respectively. An effective speed of sound in the transition zone is defined as $c_{12}(T) = c_1 + f_s c_2$. The velocity of microscopic volumes of the liquid is calculated as: $v_1 = \frac{dS_1}{dt}$. The deformation and stress are defined as: $\varepsilon_1 = \frac{dS_1}{dr}$ and $\sigma_1 = E_1 \varepsilon_1 = E_1 \frac{dS_1}{dr}$, respectively, where $E_1$ is the volumetric modulus of melt elasticity which is determined from the known ratio in the longitudinal wave: $E_1 = c_1^2 \rho_1$. The values $v_{12}$, $\varepsilon_{12}$, $\sigma_{12}$ are also calculated in the transition zone.

Consider an integral characteristic of melt stresses during the time of solidification:

$$Z_\sigma = \frac{1}{r_0} \int\limits_{t_{col}}^{t_{froz}} |\sigma_1|. \tag{6}$$

$Z_\sigma$ has a dimensionality of mechanical impedance $Z_s = c_1 \varrho_1$. This value can be considered as a stress integral during solidification related to the unit element of solidification front trajectory from the mold walls to its axis $r_0$. It also can be considered as the total value of stresses of the vibration field directed at overcoming the mechanical impedance during solidification.

It can be assumed that the integral value $Z_\sigma$ determines the effectiveness of vibration and provides a comparative characteristic for evaluating process conditions.

### 2.2. Modeling Results

Equations from (1) to (5) were solved numerically using an explicit difference scheme and a fixed grid method [22] implemented in Delphi 7.0.

The physical properties for an A356 alloy (Al-7% Si), listed in Table 1, were used in solving the problem.

**Table 1.** Physical quantities used in calculation.

| Properties | Density $\varrho$, kg/m$^3$ [23,24] | Specific Heat c, J/(kg·°C) [24,25] | Thermal Conductivity $\lambda$, W/(m·°C) [24,25] | Elastic Modulus $E$ $10^{-5}$, MPa [23] | Sound Speed $c_s$, m/s [23] | Liquidus Temperature $T_{m1}$, K [26] | Solidus Temperature $T_{m2}$, K [26] |
|---|---|---|---|---|---|---|---|
| Melt (1) | 2362 | 1177 | 98.1 | 0.52 | 4700 | 883 | 841 |
| Solid metal (2) | 2660 | 880 | 155.0 | 0.70 | 6260 | - | - |
| Steel (3) | 7800 | 462 | 50.2 | - | - | - | - |

The ranges of process parameters in the calculation were as follows:

- Frequency $f$ = 0–100 Hz
- Amplitude $A$ = 0.1–10 mm
- Initial temperature of the chill mold $T_s$ = 430–630 K
- Initial temperature of the liquid metal $T_0$ = 900–1050 K
- Radius of chill mold was 17.5 mm
- Specific heat of phase transition (latent heat of solidification) was $L$ = 429 kJ/kg [24].

The calculation showed that the temperature profile weakly depends on vibration within the given range of amplitude and frequency. Figure 2a shows the temperature in its center $T(0,t)$ without and with vibration at a frequency of 50 and 80 Hz. The movement of the solidification front $r_{s1}$ (that is, corresponding to the liquidus temperature $T_{m1}$) is nonlinear (Figure 2b).

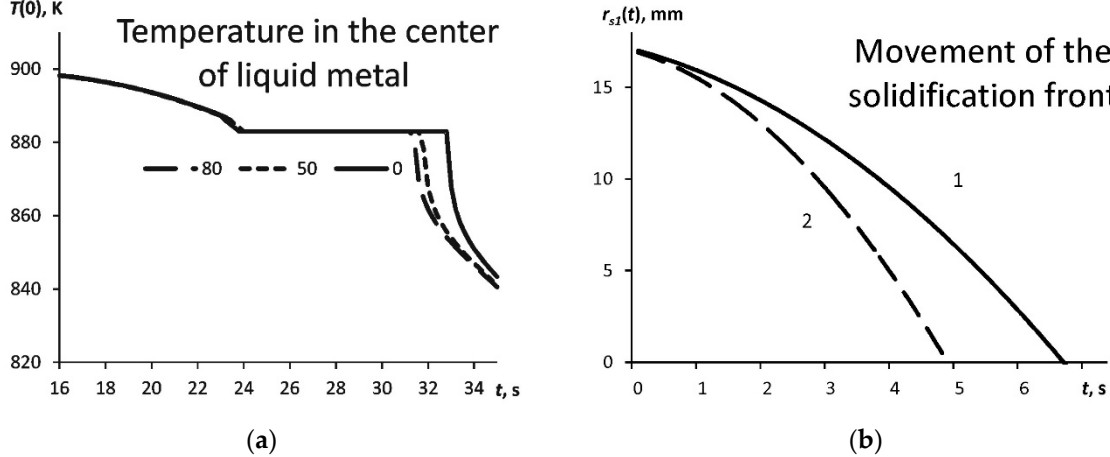

**Figure 2.** (**a**) Time dependence of the temperature in the center of liquid metal volume for different conditions: 0—without vibration, 50—vibration of $f$ = 50 Hz, 80–vibration of $f$ = 80 Hz; $T_s$ = 630 K, $T_0$ = 900 K.; (**b**) Time dependence of the movement of the solidification front from the start point $t_{col}$ for two thermal modes: 1—$T_s$ = 630 K, $T_0$ = 900 K, and 2—$T_s$ = 430 K, $T_0$ = 1050 K under $A$ = 0.5 mm, $f$ = 50 Hz.

First of all, the thermal solution allows one to define the kinetics of the process. Figure 2a shows the time-dependence of the temperature in the center of the melt volume $T(0,t)$. Vibration accelerates the process of solidification: the "plateau" on the temperature profile corresponding to the solidification becomes shorter with the vibration.

The calculation shows that the duration of alloy solidification as well as the cooling time (combined cooling to the temperature of liquidus and then to the solidus), depends on the initial temperature difference between the melt and the chill mold, and is less dependent on frequency (Figure 3a). The solidification time slightly shortens with the increasing frequency (Figure 3b). As one can see, the solidification time is short in all cases, a few seconds. However, it is during this brief period of time that the vibration has an effect on the solidification and structure formation.

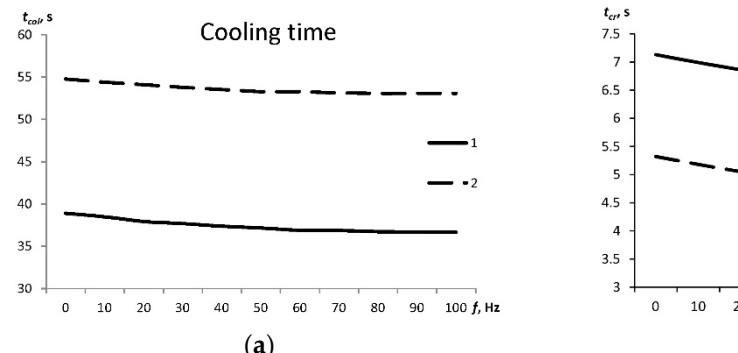
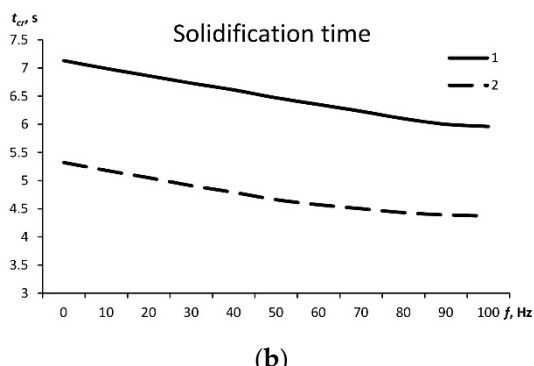

**Figure 3.** Dependencies of the cooling time (**a**) and the solidification time (**b**) on the vibration frequency: 1—$T_s$ = 630 K, $T_0$ = 900 K, and 2—$T_s$ = 430 K, $T_0$ = 1050 K; $A$ = 0.5 mm.

Figure 4 shows the dependence of the cooling time $t_{col}$ on the initial mold temperature, as well as the dependence of the solidification time on the initial melt temperature. The higher the initial temperature of the metal and that of the chill mold, the more time needed for cooling; which is the expected result.

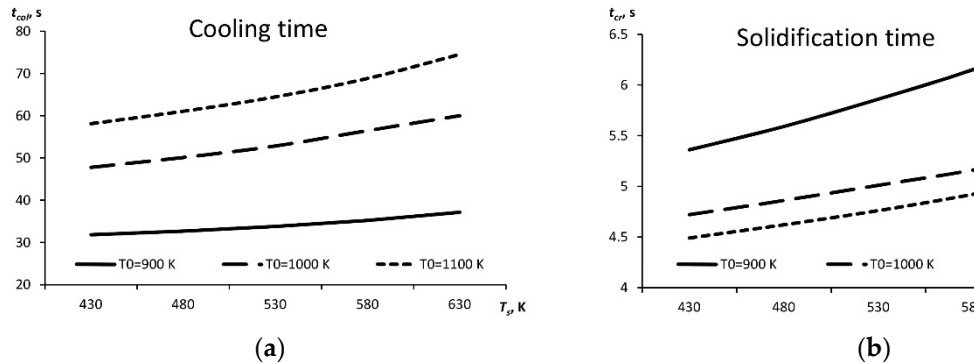

**Figure 4.** Dependence of (**a**) the cooling time before solidification and (**b**) the solidification time on the initial temperature of the chill mold $T_s$ for different starting melt temperatures $T_0$.

When analyzing the stress amplitude imposed on the semi-solid alloys, one should note that the strength of semi-solid alloys can be significantly lower than that of the solid alloy, i.e., the tensile strength of an A356 alloy decreases from 157 MPa in the solid state [27] to 4.98 MPa at 860 K and to ~ 0.01 MPa at 880 K [24]. Nevertheless, a certain critical amplitude of mechanical stress $\sigma_{min}$ that is higher than the tensile strength of the semi-solid alloy, is required to effect the fracture of growing crystals, and as a consequence, the refinement of the crystalline grains in the casting. Not only the value of $\sigma_{min}$ is important but also the operating time of these stresses during the alloy solidification (number of oscillations during solidification). A combination of those can be characterized by the value $Z_\sigma$ (Equation (6)).

Figure 5a shows the dependence of the specific stress integral $Z_\sigma$ on the vibration frequency for the two following conditions: $T_s$ = 630 K, $T_0$ = 900 K and $T_s$ = 430 K, $T_0$ = 1050 K. The calculation demonstrates that the effectiveness of the vibration grows nonlinearly with increasing the vibration frequency and saturates at a certain frequency. This effect can be explained by the reduction of the solidification time during which the vibration affects the growing crystals (see Figure 3b). It can be assumed that the fracture of growing crystals is possible only under conditions when a relative value $Z$ exceeds a certain critical value: $Z = Z_\sigma/Z_s = Z_\sigma/c_1\rho_1 > Z_{cr}$.

The dependence of this integral stress characteristic on amplitude is, however, linear and the values become substantial at the frequencies above 50 Hz (Figure 5b).

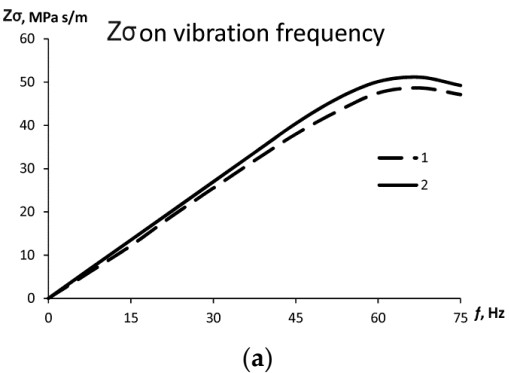
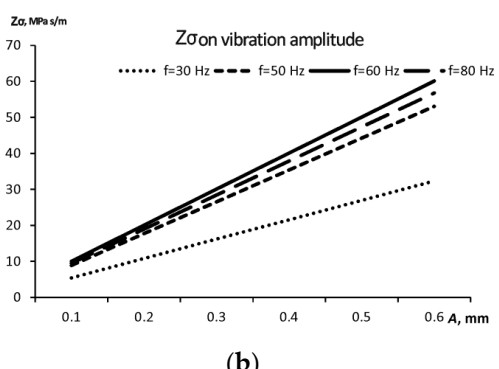

|(a)|(b)|

**Figure 5.** (**a**) The dependence of the specific stress integral on the vibration frequency for two thermal conditions: 1—$T_s$ = 630 K, $T_0$ = 900 K, and 2—$T_s$ = 430 K, $T_0$ = 1050 K; $A$ = 0.5 mm. (**b**) The dependence of the specific stress integral on the vibration amplitude for the four values of frequency.

Thus, according to the proposed approach, the vibration of liquid and solidifying metal causes the following effects:

1. Acceleration of cooling and solidification as a result of convective heat transfer regardless of little change in the temperature profile.
2. The generation of mechanical stresses in the solid–liquid zone that can potentially fracture (fragment) the growing crystals.

Considering that the vibration stresses operate on growing crystals in a particular short period of solidification, the time integral characteristic of mechanical stresses can be suggested as a measure of the effectiveness of vibration. The higher this value, the greater the effect of vibration stress on crystals. The integral characteristic of mechanical stresses increases linearly with rising amplitude but grows nonlinearly with rising vibration frequency. It is necessary to consider the time and mechanical conditions under vibration treatment of an alloy during solidification as:

- Vibration affects the structure from the beginning of solidification and until its end.
- The higher the vibration amplitude, the higher the vibration effect (it grows linearly).
- There is the optimal vibration frequency at which this effect is the highest. This frequency equals 60 Hz for parameters used in the calculation (Table 1).

## 3. Experimental Verification of the Mathematical Model by Casting of an Aluminum Alloy with Vibration

### 3.1. Experimental Procedures

A vibrating table EV 341-07 (PC "Lighthouse YF", Yaroslavl, Russia) was used to carry out the experiment to evaluate the vibration processing of solidifying melt. The scheme of the experimental setup is shown in Figure 6. Studies were performed on an A356 aluminum alloy.

The experimental technique for vibration treatment of the aluminum melt was as follows. The A356 aluminum alloy was placed in a crucible inside an electrical melting furnace (Nabertherm GmbH, Lilienthal, Germany) at a furnace temperature of 1073 K. A steel chill mold (with an internal diameter of 35 mm, wall thickness of 135 mm, height of 200 mm), preheated to a given temperature 430 K, was fixed on a vibrating table that oscillated in a mode of horizontal vibration (the same condition as used in the model described above). Then, using a holding device we removed the crucible from the furnace and poured the liquid alloy in the preheated chill mold at a temperature of 973 K (700 °C). The fixed vibration frequencies were 50, 60 and 80 Hz in order to cover the range around the maximum in Figure 5a. The vibration amplitude was varied in the range from 0.38 to 0.53 mm (angle of an off-center block is from 10° to 20°, the link between amplitude and the angle θ [rad] is expressed as follows: $A = \frac{\pi - \theta}{\pi} A_{\max}$, where $A_{max}$ = 0.56 mm). The vibration continued for

1–2 min after pouring until the complete solidification of the melt. A reference experiment without vibration (with the other parameters the same) was conducted to assess the effectiveness of vibration (grain structure and density were chosen as the metallurgical indicators).

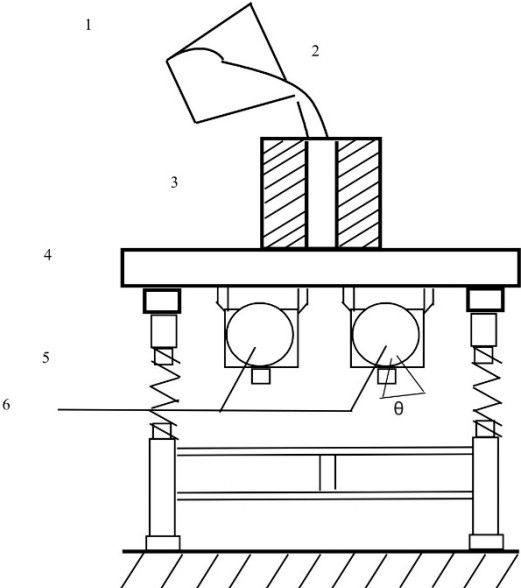

**Figure 6.** Scheme of the experimental setup: 1—crucible; 2—liquid metal; 3–chill mold; 4—table; 5—springs; 6—vibrators.

*3.2. Analytical Techniques*

The structure of the obtained materials was investigated using optical microscopy (Olympus GX-71, Tokyo, Japan) and scanning electron microscopy (Philips SEM-515, Hillsboro, Oregon, USA). A universal testing machine Instron 3369 was used for mechanical tensile testing of the obtained alloys. Three samples were tested for each condition. The density of the obtained alloys was estimated by a hydrostatic weighing method (using Archimedes' principle).

Samples were pre-polished on a circular grinding machine (Buehler, Lake Bluff, IL, USA) and then subjected to mechanical polishing. Samples were etched using a 5% HBF4 acid solution to identify the microstructure. Mechanical tensile tests are carried out with a tensile test machine Istron 3369 (Norwood, MA, USA). The samples were cut in the form of plates (length of the working part 25 mm, width 6 mm, thickness 2 mm, rounded radius 18 mm) using electroerosion cutting of aluminum alloy. Tensile testing are conducted at a rate of 0.2 mm/min until a fracture occurs. Test results were obtained in the form of "stress-strain" diagrams from which the values of yield strength, tensile strength and maximum elongation were found.

## 4. Results and Discussion

At a frequency of 60 Hz and the angle of off-center block $\theta = 20°$ (amplitude of 0.53 mm), the density of samples cut from the lower part of an ingot increases in comparison with the density obtained at 50 Hz (Table 2).

Increasing the density of samples may be linked to the facilitated gas (hydrogen) release during melt vibration that decreases gas and shrinkage porosity [5,28]. A further increase in the vibration frequency up to 80 Hz leads to a decrease in the density of an ingot. A similar trend is observed at the angle of off-center block $\theta = 10°$ (amplitude of 0.38 mm), i.e., the increased frequency 80 Hz slightly reduces the density of the samples.

These measurements correlate well with the modeling (Figure 5a): the integral of mechanical stresses increases with the vibration frequency up to 60 Hz and then slightly decreases to 80 Hz. This is

due to the reduction in the solidification time with increasing frequency as a result of intensified thermal processes in the melt, which consequently reduces the effective time of vibration and for degassing.

The observation of the macrostructure of the obtained A356 ingots without (Figure 7a) and with vibration (Figure 7b) shows that vibration processing (frequency of 60 Hz and amplitude of 0.5 mm) leads to a significant grain refinement.

**Table 2.** Measurements of the density of A356 aluminum alloy.

| No. of Sample | Frequency, Hz | Amplitude, mm (θ, °) | Density, g/cm³ |
|:---:|:---:|:---:|:---:|
| 1<br>2<br>3<br>4 | 50 Hz | | 2.70 ± 0.02 |
| 1<br>2<br>3<br>4 | 60 Hz | 0.53 mm (20 °) | 2.72 ± 0.03 |
| 1<br>2<br>3<br>4 | 80 Hz | | 2.69 ± 0.03 |
| 1<br>2<br>3<br>4 | 50 Hz | | 2.69 ± 0.04 |
| 1<br>2<br>3<br>4 | 60 Hz | 0.38 mm (10 °) | 2.70 ± 0.02 |
| 1<br>2<br>3<br>4 | 80 Hz | | 2.69 ± 0.02 |

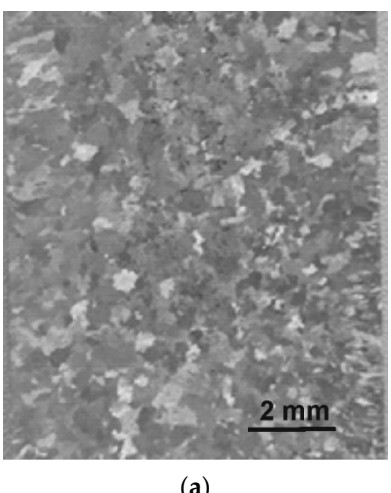 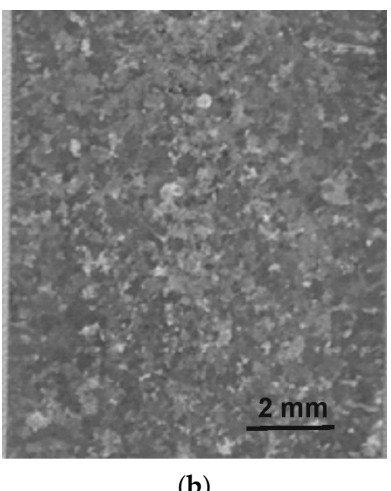

(**a**)  (**b**)

**Figure 7.** Macrostructure of an A356 ingot: (**a**) obtained without vibration; (**b**) obtained with vibration during solidification (frequency of 60 Hz and amplitude of 0.5 mm).

The microstructures of the A356 alloy samples (Figure 8) show that significant structural changes occurred during the vibration. The average grain size reduces from 208 μm in the alloy produced without vibration to 89 μm for the alloy cast with vibration (Figure 9).

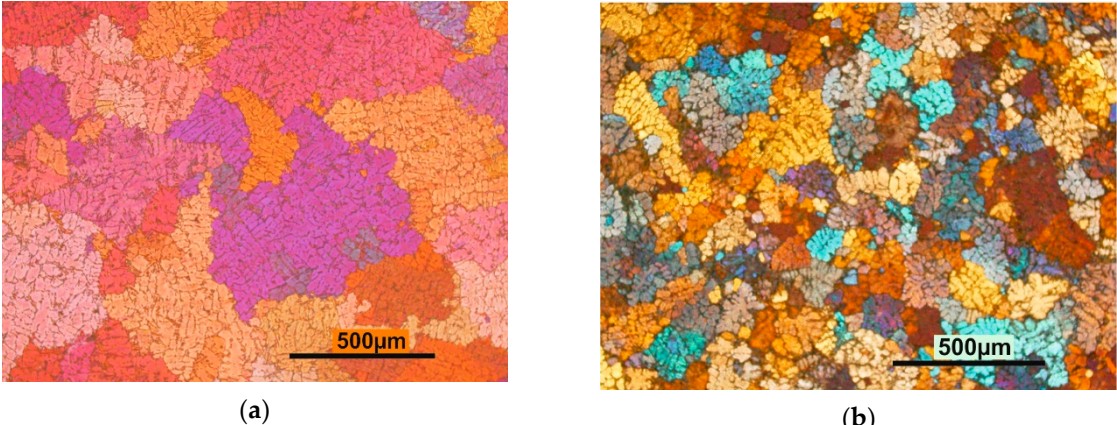

(**a**)　　　　　　　　　　　　(**b**)

**Figure 8.** Microstructure of an A356 aluminum alloy: (**a**) without vibration; (**b**) after vibration during solidification (frequency of 60 Hz and amplitude of 0.5 mm).

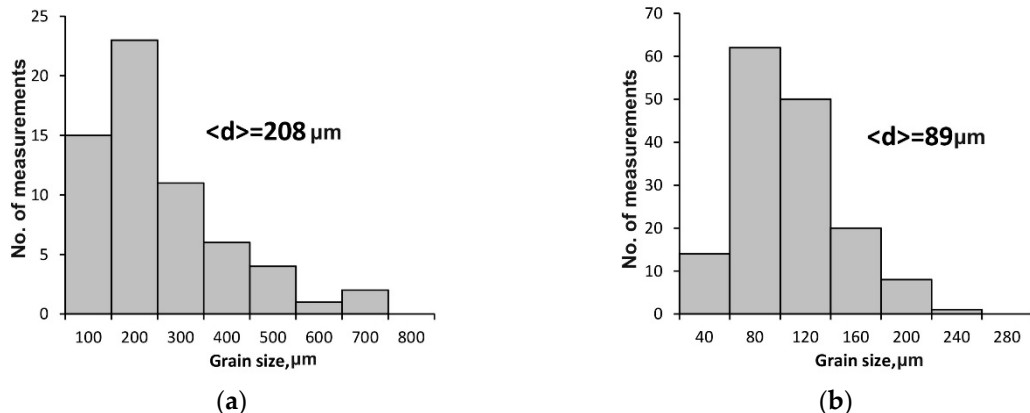

(**a**)　　　　　　　　　　　　(**b**)

**Figure 9.** Grain size distribution in an A356 aluminum alloy (**a**) without vibration and (**b**) after vibration during solidification (frequency of 60 Hz and amplitude of 0.5 mm).

Based on the obtained experimental and theoretical results, we can conclude that the optimal frequency for vibration processing for the given mold geometry and the range of parameters is 60 Hz. The calculations also showed that the higher the amplitude, the better the result of the treatment. The experimental study showed a similar trend with the maximum density of the samples obtained at the highest amplitude used (0.53 mm, angle of 20°).

The improvement of the structure in the castings obtained with vibration is related to intensive heat transfer in the melt. The vibration energy is spent on the fragmentation of dendritic branches, and this process results in grain multiplication. The alloy structure obtained by vibration shows the presence of very small grains (Figure 8b) which represent preserved dendrite fragments.

Mechanical testing of the as-cast A356 aluminum alloy shows that yield strength $\sigma_{0.2}$ increases from $67 \pm 6$ to $121 \pm 7$ MPa after vibration of the melt with a frequency of 60 Hz and an amplitude of 0.53 mm compared to the sample without vibration. This result can be associated with the decrease in the grain size and porosity and the increased density of the samples. Meanwhile, the tensile strength remains unchanged at the level $\sigma_B = 182 \pm 7$ MPa and the elongation $\delta$ remains at the level $\delta = 3.4 \pm 0.2\%$ (Table 3, Figure 10).

**Table 3.** Mechanical testing of A356 aluminum alloy.

| Frequency, Hz | $\sigma_B$, MPa | $\delta$, % |
|---|---|---|
| Without vibration | $150 \pm 7$ | $2.1 \pm 0.1$ |
| 50 | $160 \pm 6$ | $2.3 \pm 0.1$ |
| 60 | $182 \pm 7$ | $3.4 \pm 0.2$ |
| 80 | $149 \pm 6$ | $2.1 \pm 0.1$ |

**Figure 10.** Stress-strain diagrams of the A356 aluminum alloy without vibration and after vibration during solidification (frequency of 60 Hz and amplitude of 0.5 mm).

Our results agree well with the work of Murakami et al. [17] who studied the effect of vibration frequency, acceleration amplitude, velocity amplitude, and displacement amplitude on the size and shape of the grains in an JIS AC4CH aluminum alloy (A356 analog). They used the frequencies of 10, 20, 50,100,150 and 200 Hz and found that the smallest grains (132 μm) were obtained at a frequency of 50 Hz and an amplitude of 0.49 mm, which is close to our result. However, their work did not cover the frequency range of our interest, i.e., 50 to 80 Hz.

It is important to note that in the frequency range 50–80 Hz and amplitudes up to 1 mm there are no conditions for cavitation and turbulent fluid flow [15]. At the same time, there are no conditions for entrapment of atmospheric gases and oxide films, which leads to extremely undesirable porosity in the ingot. When implementing more intensive and high-frequency process conditions, both the positive effects of turbulence and cavitation (better mixing, higher stresses, efficient degassing) and the negative ones (entrapment of gases and oxides leading to pores and cavities in the metal) should be considered.

## 5. Conclusions

In this study we showed that the experimentally observed improving of an as-cast structure at low-frequency and low-intensity vibration can be reasonably described by the thermal model (Stefan problem with mechanical vibration).

Optimal conditions for low-frequency and low-intensity vibration of the melt in a cylindrical chill mold were determined by theoretical calculation and verified by experimental testing, that is, a vibration frequency of about 60 Hz and an amplitude of about 0.5 mm. Under such conditions, there is a significant decrease in the grain size in the ingot as well as improved soundness (density) of the as-cast metal. The yield strength increased by ~80% as compared to the ingot cast without vibration.

The optimal conditions within the thermal model are explained by the reduction of the solidification time with increasing frequency on the one hand, and by the growing integral mechanical stresses during solidification on the other hand. The specific integral of mechanical stresses in the melt subjected to vibration during solidification can be considered as a measure of the effectiveness of vibration within the thermal approach for the process description.

Theoretical calculation allows one to optimize the time of the vibration effect. The vibration has no practical value if applied above the liquidus temperature. Moreover, it can act as a source of the undesirable phenomena of gas capture. The vibration should be started just at the beginning of solidification and finish with the complete metal solidification.

The mathematical model presents a cylindrical mold with horizontal vibration. The model does not take into account other geometric variations, nor the vertical component of vibration. In future, it would be useful to simulate other conditions of metal solidification with vibration.

**Author Contributions:** Conceptualization, A.V. and D.E.; Data curation, V.D. and D.E.; Formal analysis, O.K.; Funding acquisition, A.V.; Investigation, D.E.; Methodology, O.K. and P.D.; Project administration, A.V.; Resources, V.D.; Software, O.K.; Supervision, A.V.; Validation, M.K. and V.D.; Writing—original draft, O.K.; Writing—review & editing, M.K. and P.D.

**Funding:** This research was funded by Russian Science Foundation, grant number 17-13-01252.

**Conflicts of Interest:** The authors declare no conflict of interest.

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
