# Peer review of "Optimizing the Conditions of Metal Solidification with Vibration"

_metals, doi:10.3390/met9030366_

Round 1
Reviewer 1 Report
Dear Authors,
Following are some remarks to the work:
1. Please add full address of authors affiliation: department, full address, ZIP, etc.
2. In literature review I didn’t find a recent publication in simultaneous influence of vibration and incorporation of nanoparticles into Al alloy which is relevant to the work: V. Selivorstov, Y. Dotsenko, K. Borodianskiy, Influence of Low-Frequency Vibration and Modification on Solidification and Mechanical Properties of Al-Si Casting Alloy, Materials 10 (5) (2017) 560-570. DOI: 10.3390/ma10050560.
3. Experimental procedure. You have mentioned section 2.1 and no 2.2, I guess you should add section 2.2: Analytical techniques.
4. In experimental technique please add the etchant composition you have used for optical microscopy investigations and parameters you have used for tensile tests by Instron.
5. You have made conclusions that 60Hz is the optimal frequency should be used in the process (lines 270-272). Unfortunately you have not showed any comparative results for other frequencies. Please explain or add results obtained by other frequencies.
6. You must add tensile test plots for investigated alloys compare them with the alloy that poured without vibration at the same conditions, and then discuss the obtained results.
Author Response
Point 1: Please add full address of authors affiliation: department, full address, ZIP, etc.
Response 1: Full address of authors affiliation were added.
Point 2: In literature review I didn’t find a recent publication in simultaneous influence of vibration and incorporation of nanoparticles into Al alloy which is relevant to the work: V. Selivorstov, Y. Dotsenko, K. Borodianskiy, Influence of Low-Frequency Vibration and Modification on Solidification and Mechanical Properties of Al-Si Casting Alloy, Materials 10 (5) (2017) 560-570. DOI: 10.3390/ma10050560.
Response 2: A ref. 8 was changed to more relevant ref: “Selivorstov, V.; Dotsenko, Y.; Borodianskiy, K. Influence of low-frequency vibration and modification on solidification and mechanical properties of Al-Si casting alloy. Materials, 2017, 10( 5), 560.”
Corresponding phrase was inserted to page 1: “Study of the mechanical properties of A356 alloy after vibration treatment with the frequencies of 100 Hz and 150 Hz showed the improved tensile and yield strengths by 20% and 10% respectively, but the use of vibration frequencies of 200 Hz caused the formation of a high porosity microstructure and caused major defects [8].”
Point 3: Experimental procedure. You have mentioned section 2.1 and no 2.2, I guess you should add section 2.2: Analytical techniques.
Response 3: Section “Analytical techniques” was added (comment 3 of reviewer 1), page 8.
Point 4: In experimental technique please add the etchant composition you have used for optical microscopy investigations and parameters you have used for tensile tests by Instron.
Response 4: We added the text to the section “Analytical techniques”:
Samples were pre-polished on a circular grinding machine and then subjected to mechanical polishing. Samples were etched using a 5% HBF4 acid solution to identify the microstructure. Mechanical tensile tests are carried out with a tensile test machine Istron 3369. The samples were cut in the form of plates (length of the working part 25 mm, width 6 mm, thickness 2 mm, rounded radius 18 mm) using electroerosion cutting of aluminum alloy. Tensile test are conducted at a rate of 0.2 mm/min until a fracture occurs. Test results were obtained in the form of diagrams "stress-strain" from which the values of yield strength, tensile strength and maximum elongation were found.
Point 5: You have made conclusions that 60Hz is the optimal frequency should be used in the process (lines 270-272). Unfortunately you have not showed any comparative results for other frequencies. Please explain or add results obtained by other frequencies.
Response 5: First of all, we carried out a comparison for the density of samples after vibration processing at three frequencies. The maximum density was reached with a frequency of 60 Hz (for two values of amplitude, table 2). Then we compared a microstructure and the grain size in samples after vibration for the "best" frequency and without vibration processing. More detailed research on structure of samples after vibration processing with a frequency of 10, 20, 50,100,150 and 200 Hz of AC4CH aluminum alloy are obtained in work [Murakami, Y.; Miwa, K.; Kito, M.; Honda, T.; Kanetake, N.; Tada, S. Effects of Mechanical Vibration Factors on Size and Shape of Solid Particles in JIS AC4CH Aluminum Alloy Semi-Solid Slurry. Material Transactions, 2016, 57 (2), 163-167]. They found the extremum for density of samples and grain size on the frequency of 50 Hz. It is close to our results.
We added some data for mechanical testing of samples without vibration processing and after vibration with 3 frequencies(50, 60, 80 Hz) – Table 3.
Point 6: You must add tensile test plots for investigated alloys compare them with the alloy that poured without vibration at the same conditions, and then discuss the obtained results.
Response 6: We added the tensile test plot for the case without vibration processing and with the frequency of 60 Hz (Fig. 10).
Reviewer 2 Report
The authors solved the Stefan problem of a solidification process in a cylindrical chill mold with vibration numerically. As a result, they obtained dependence of this value on the vibration frequency and amplitude. They verified the numerical results with the experiment by using a A356 alloy. The optimum conditions for the numerical simulation and experiments were almost the same.
The article shows new results, which are worth for publication.
Few things should be corrected before publication.
Fig. 7: there are no scale bars
Fig. 8: scale bars are hardly seen
Indices: The authors do not use the same style for indices throughout the paper. That should be corrected.
Spaces in the equations: There are many examples, a typical one is in the caption for Figure 2.
The authors wrote: T0=900 K, without the index (0), without spaces between variable, equal sign and value (T0 = 900 K). Everything should be carefully checked before the next submission.
Author Response
Point 1: Fig. 7: there are no scale bars.
Response 1: Scale bars were added.
Point 2: Fig. 8: scale bars are hardly seen.
Response 2: Scale bars were made brighter.
Point 3: Indices: The authors do not use the same style for indices throughout the paper. That should be corrected.
Response 3: Corrected.
Point 4: Spaces in the equations: There are many examples, a typical one is in the caption for Figure 2. µm.
Response 4: Spaces were added to the equations.
Point 5: The authors wrote: T0=900 K, without the index (0), without spaces between variable, equal sign and value (T0 = 900 K). Everything should be carefully checked before the next submission.
Response 5: Were corrected, were checked.
Reviewer 3 Report
The idea behind the manuscript is interesting but authors had made some mistakes and some plots are not shown and sections numbering is wrong. Manuscript has two sections “2” and two “2.1” subsections under the “first 2 section”.
Line 106: why use “-“ symbol instead of “corresponds to”?
Line 110: Why use “melt” instead of “liquid phase”.
Lines 125-128: A 4 lines sentence is too long. Split it into 2 or 3 shorter sentences.
Lines 131: Sub-section 2.1 was previously used.
Line 136: Elastic modulus units are not right. 10^-5 MPa? That means tens of Pascals? Here, in this table authors use “.” As decimal point? Or no? For densities “.” are used for thousands and in the rest of the talble for decimals? According to the SI standards in English language “.” is use as decimal separator and “,” is used for thousands.
Line 146: Where is the figure 2a where you can se the profile for 120 seconds? Figure 2 does not corresponds to this.
Line 156: Fig 2b do not show what you said in text.
Line 173: Use points for figure 4 instead of lines as you calculate the t for some discrete values of temperature.
Line 184 Figure 5 is not correct as it is a duplicate of figure 4.
Line 212… section 2 again?
Line 214… subsection 2.1? There are no more subsection in this section so the 2.1 must be removed.
Line 240: Section 4
Line 244: Table units for angle is indicated as “rad” but then degrees were used. In second part of the table again 20º was used but in text (line 249) it was referred as 10º. That must be corrected.
It is no clear the relation between the model and the experimental results. On one side you predict temperatures, times and solid-front speed and in the other one density grain size… how you probe that you model is correct with those results?
Author Response
Point 1: The idea behind the manuscript is interesting but authors had made some mistakes and some plots are not shown and sections numbering is wrong. Manuscript has two sections “2” and two “2.1” subsections under the “first 2 section”.
Response 1: Was corrected.
Point 2: Line 106: why use “-“ symbol instead of “corresponds to”?
Response 2: Was corrected.
Point 3: Line 110: Why use “melt” instead of “liquid phase”.
Response 3: Corrected.
Point 4: Lines 125-128: A 4 lines sentence is too long. Split it into 2 or 3 shorter sentences.
Response 4: Was split into 2 sentences.
Point 5: Lines 131: Sub-section 2.1 was previously used.
Response 5: Corrected.
Point 6: Elastic modulus units are not right. 10^-5 MPa? That means tens of Pascals? Here, in this table authors use “.” As decimal point? Or no? For densities “.” are used for thousands and in the rest of the talble for decimals? According to the SI standards in English language “.” is use as decimal separator and “,” is used for thousands.
Response 6: Thank you, corrected. E 10-5 , MPa (Comma on the right place); “,” is used for thousands and “.” for decimals.
Point 7: Line 146: Where is the figure 2a where you can se the profile for 120 seconds? Figure 2 does not corresponds to this.
Response 7: We can see only the temperature in its center, indeed. The phrase “the temperature distribution at t=120 s in the metal volume from the mold center to its wall and” was deleted.
Point 8: Line 156: Fig 2b do not show what you said in text.
Response 8: Corrected in the text under the figure (Fig 2a is right).
Point 9: Line 173: Use points for figure 4 instead of lines as you calculate the t for some discrete values of temperature.
Response 9: Sorry, but I do not agree, the cooling time is a continuous function of the temperature of Ts but for three different reference temperatures of T0 (three curves).
Point 10: Line 184 Figure 5 is not correct as it is a duplicate of figure 4.
Response 10: Corrected figures were inserted.
Point 11: Line 212… section 2 again?
Response 11: Corrected.
Point 12: Line 214… subsection 2.1? There are no more subsection in this section so the 2.1 must be removed.
Response 12: Corrected.
Point 13: Line 240: Section 4
Response 13: Corrected.
Point 14: Line 244: Table units for angle is indicated as “rad” but then degrees were used. In second part of the table again 20º was used but in text (line 249) it was referred as 10º. That must be corrected.
Response 14: Corrected.
Point 15: It is no clear the relation between the model and the experimental results. On one side you predict temperatures, times and solid-front speed and in the other one density grain size… how you probe that you model is correct with those results?
Response 15: In the result of solving of Stefan problem, we propose the specific integral of mechanical stresses in the melt subjected to vibration during solidification as a measure of the effectiveness of vibration within the thermal approach for the process description. This integral has an extremum at a frequency of about 60 Hz. The best results for the structure and mechanical properties of samples we saw in an experiment also at 60 Hz. So we make a conclusion about the adequacy of the model.
Round 2
Reviewer 1 Report
Dear Authors,
Thank you for considering all remarks I have mentioned.
Reviewer 3 Report
Authors follow reviewer's comments and clarified all comments. I think that in its actual form paper is publishable.